# Challenges in Assessing Repellency via the Behavioral Response by the Global Pest *Tribolium castaneum* to Protect Stored Grains

**DOI:** 10.3390/insects15080626

**Published:** 2024-08-20

**Authors:** Leslie C. Rault, William R. Morrison, Alison R. Gerken, Georgina V. Bingham

**Affiliations:** 1Department of Entomology, University of Nebraska-Lincoln, 103 Entomology Hall, Lincoln, NE 68583, USA; lrault2@unl.edu; 2United States Department of Agriculture, Agricultural Research Service, Stored Product Insects and Engineering Research Unit, 1515 College Ave., Manhattan, KS 66502, USA; william.morrison@usda.gov (W.R.M.III); alison.gerken@usda.gov (A.R.G.)

**Keywords:** wind tunnel, Ethovision, IPM, semiochemicals, Coleoptera, essential oils

## Abstract

**Simple Summary:**

This study addresses global food security by highlighting the challenge of protecting food losses from insect damage. To find effective low-risk ways to manage these pests, we assessed the avoidance of the red flour beetle, a common pest in stored grains, to essential oils. Through various experiments in the laboratory, we investigated the beetles’ behavioral response to different scented oils. While no definitive repellent was identified, this study sheds light on factors influencing insect behavior and the need for further research in behaviorally based management tactics such as avoidance. Ultimately, this work aims to develop strategies to protect stored grains, contributing to global food security efforts.

**Abstract:**

Background: Food security is an increasingly pressing global issue, and by 2050, food production will not be sufficient to feed the growing population. Part of global food insecurity can be attributed to post-harvest losses, including quantity and quality losses caused by stored-product pests like insects. It is thus timely to find management strategies to mitigate these losses and counteract food insecurity. The red flour beetle, *Tribolium castaneum* (Herbst) (Coleoptera: Tenebrionidae), a global stored-product pest with a wide range of food sources, was used in this study to assess repellency to a selection of essential oils. Methods: Multiple behaviorally relevant methods were used to determine the efficacy of the essential oils in assays to pinpoint the most promising repellents. Experiments were used to assess individual and group behaviors with or without airflow and examined the behavioral variation in distance moved and the time spent away from the oil. Results: It was found that exposure to essential oils and conditions of experimentation considerably influenced *T. castaneum*’s behavioral response, but a clear candidate for repellency could not be chosen based on the collected data. Conclusions: Follow-up research is needed to pinpoint repellents for integrated pest management practices to protect grains from stored-product pests and to justify their use in and around commodities.

## 1. Introduction

The combination of population growth and food availability will intensify the strain on our agricultural systems worldwide by 2050. Food security will be dramatically impacted if increased yields and quality needs are not met through improved food and agricultural production management practices. Currently, between 702 and 828 million people worldwide do not have access to food to fulfill their nutritional needs. This trend will only grow, and is exacerbated by conflicts and the SARS-CoV2 pandemic [1,2,3,4,5]. One-third of food commodities produced are lost or wasted post-harvest [6]; understanding the complex interrelated factors that lead to severe crop loss though the post-harvest food value chain—from farm to fork—is critical. Integrated partnerships are needed to evaluate all areas of the food value chain to devise targeted innovative, science-based discoveries to protect production gains already achieved and support informed, practical decisions for continued growth. One crucial way to protect commodities and enhance their longevity in storage is through improving pest control [3,6,7]. It has been estimated that 9–20% yield loss is due to insect pest damage after harvest, significantly impacting global nutritional security and food safety [6,7,8]. Increasing crop production prior to harvest usually requires significantly more resources, particularly for producers, than targeted strategies for reducing losses post-harvest [6,7,9], with economic costs from post-harvest losses compounded after substantial investments in production and processing have already been made and cannot be recouped [10]. Thus, improved protection of post-harvest production protects labor and capital investments already made, in addition to contributing to food security.

The US post-harvest commodity market is a multi-billion dollar economy, and US food loss has been valued at USD 161 billion annually [11]. Estimates are that 10–20% of these post-harvest commodities decline in food quality and nutrition before reaching the consumer [9,12,13,14]. In addition, stored-product insects and improper storage have also been associated with antibiotic-resistant and virulent pathogens such as mycotoxins that have been major sources of disease outbreaks in our food supply [15]. Mycotoxin contamination is a long-standing global threat to food safety and food security. Aflatoxin is one type of mycotoxin produced by fungi, primarily *Aspergillus flavus*, often found in maize and peanuts and carried by insects [16,17,18,19,20]. The United States Centers for Disease Control and Prevention (CDC) reported that 4.5 billion people globally have been chronically exposed to aflatoxin [21], and 15,000 to 25,000 liver cancer cases are associated with chronic aflatoxin exposure in Asia and sub-Saharan Africa every year [22].

Options for protecting post-harvest commodities are shrinking due to insecticide resistance, consumer demands, and increasing regulatory restrictions. Continued success in insect management will require innovative, holistic approaches [13,23,24]. Given the high economic losses associated with infestations and the difficulty in eliminating infestations after they become established, most of the efforts in IPM programs against stored-product pests should be directed towards preventing infestations from occurring [25,26], and implementing resistance management programs using chemical control options with alternative modes of action to allow for proper pesticide stewardship.

Overall, stored-product insect pests are a cause for concern for their contribution to food insecurity worldwide. The red flour beetle, *Tribolium castaneum* (Herbst) (Coleoptera: Tenebrionidae), is an external-infesting feeder, classified as a major pest, capable of rapidly causing extensive and significant damage to stored products [27]. It infests cereal grains and their products, such as wheat or rice flours [28,29]. *T. castaneum* is a model insect and pest, often used for genetics research [30] and overall fundamental and applied research [31]. Publications are abundant on management practices and their efficacy against *T. castaneum*, encompassing traditionally applied synthetic insecticides treatments, but also long-lasting insecticidal nets, insecticide-treated or -incorporated bags and hermetic bags for grain storage, or attractive toxic sugar bait stations (ATSB) [32,33,34,35,36,37,38,39]. Various studies have also characterized *T. castaneum* stress responses when exposed to insecticides [40,41,42,43,44], as well as resistance to insecticidal compounds [45,46,47,48].

With insecticide resistance being a worldwide concern, alternative modes of action and strategies need to be implemented for successful integrated pest management (IPM) programs. Among the tools for the management of stored-product pests, including *T. castaneum*, semiochemical-mediated behaviorally based strategies are a sub-suite of strategies that include mating disruption and attract-and-kill, but also repellency [49]. The key feature is that externally applied semiochemicals manipulate the populations of stored-product insects to protect the commodities of interest. One way that repellency can be achieved is through the use of essentials oils and plant extracts, which have been found to have potential for repellency and some level of toxicity [44,50,51,52]. Essential oils are considered to be eco-friendly approaches for pest management, mainly in juxtaposition to synthetic insecticides [33,53,54], and have shown potential for synergy with phosphine for *T. castaneum* [55] and as fumigants for other stored-product insects [25] or as synergists with pyrethroids in other insects, such as mosquitoes [56,57,58].

However, repellency testing requires assessing behavior, which can be difficult and subject to a large amount of environmental variation [59]. In this study, we set up multiple types of experiments, aiming at identifying candidate repellents that could be used in IPM programs to manage *T. castaneum* populations in stored-product warehouses and small farms. Although some of the tested essential oils showed promise, results were inconsistent from one experiment to the next. This study shows how critical data interpretation is in the context of adding additional tools to an IPM program in stored products.

## 2. Materials and Methods

### 2.1. Insects

The described laboratory tests were conducted on mixed-sex adult *T. castaneum*. All beetles were collected from laboratory colonies maintained in glass mason jars under a 12:12 light: dark cycle at 28 °C ± 2 °C and in relative humidity of 65% ± 5% in environmental chambers (Percival Scientific Inc., Perry, IA, USA). *T. castaneum* were continuously fed on organic flour (unbleached Flour, King Arthur, White River Junction, VT, USA). The insects used to establish the colonies were provided by the Center for Grain and Animal Health Research (CGAHR), USDA-ARS, Manhattan, KS, and were non-resistant (wild type) insects collected from eastern KS in 2012, and were maintained on a mix of 95% organic flour and 5% brewer’s yeast.

### 2.2. Essential Oils and Other Chemicals

The following 100% pure therapeutic grade essential oils tested were purchased from Piping Rock (Ronkonkoma, NY, USA); we tested bergamot (*Citrus bergamia* Risso), citronella (*Cymbopogon winterianus* Jowitt ex Bor), eucalyptus (*Eucalyptus globulus* Labill.), lime (*Citrus aurantifolia* [Christm.] Swingle), verbena (*Verbena officinalis* L.), wintergreen (*Gaultheria procumbens* L.), cedarwood (*Juniperus ashei* J. Buchholz), lemongrass (*Cymbopogon flexuosus* [Nees ex Steud.] W.Watson), calendula (*Calendula officinalis* L.), birch tar (*Botula alba* Roth), rosemary (*Rosemarinus officinalis* Spenn.), black pepper (*Piper nigrum* L.), and peppermint (*Mentha piperita* L.). Additionally, garlic (*Allium sativum* L.) 100% pure therapeutic grade essential oil was purchased from Healing Solutions (Phoenix, Arizona) on Amazon in September 2018. Three other compounds, known for their potential repellent effects on insects, were used: N,N-Diethyl-m-toluamide (DEET) at 25% (Thermo Fisher Scientific, Waltham, MA, USA); methylbenzoquinone or MBQ (Sigma Aldrich, St Louis, MO, USA), a compound naturally released by *T. castaneum* in response to crowding [60]; and transfluthrin or TF (Sigma Aldrich, St Louis, MO, USA), a pyrethroid insecticide that has a repellent effect on mosquitoes [61]. Concentrations of these two compounds were prepared in acetone at 40 mg/mL (the equivalent of the amount released by 2000 *T. castaneum)* for MBQ and 1 mg/mL for TF [60,61]. Acetone was used as a solvent for these three compounds and was also purchased from Thermo Fisher Scientific. Storgard, a broad-spectrum oil-based kairomone food attractant (Trécé Inc., Adair, OK, USA), was used in the large wind tunnel studies as positive control for attraction.

### 2.3. Essential Oil Screening Experiments

We evaluated the behavior of *T. castaneum* when exposed to 2–10 µL (depending on the assay) of essential oils to determine if they could be potential repellents of stored-product insect pests. The specific oils tested were based on the results of the sequential experiments described below. In total, the experiments consisted of the following: (1) a small wind tunnel assessed for individual and group responses; (2) a large wind tunnel assay, assessed for individual responses only; (3) behavioral tracking in a Petri dish for individuals only; and (4) a release–recapture assay for group response only. These assays capture a range of responses from close- to long-range and account for some behavioral variability resulting from group effects. All of the results were analyzed with GraphPad Prism version 10.3.0 for Windows (GraphPad Software, La Jolla, CA, USA), and analyses were specific to each assays’ design. Analyses specific to each assay are detailed below.

#### 2.3.1. Small Wind Tunnel

We used a small wind tunnel, as has been discussed prior [62], to assess the response of *T. castaneum* to key essential oils and stimuli. Briefly, the wind tunnel comprised an electric fan measuring 12 cm × 12 cm × 3 cm L:H:W that pushed ambient air through a charcoal filter (Figure 1), then straightened air flow through a metal grate and condensed the flow to a space measuring 12 cm ×  5 cm over a 26.5 cm distance within a nonvolatile steel encasement. The wind tunnel produced a laminar flow of air speed of 0.76–0.91 m/s. A release arena of 21.5 cm × 21.5 cm was placed 18 cm downwind of the wind tunnel, thereby locating odor sources 9 cm downwind of the small wind tunnel and 9 cm upwind of the release arena. Each compound was added by pipetting 10 µL neat onto a plastic square (4 cm × 4 cm) placed halfway between wind tunnel and release arena. The distribution for each oil was also compared to a negative control consisting of 10 µL of distilled water. There were two tests performed with this apparatus, as described below.

##### Individual Exposure

Individual *T. castaneum* were added to the center of the release arena at the beginning of the assay (Figure 1). A total of n = 30 replicate *T. castaneum* were exposed successively to each oil. Oils tested here were the following: citronella, cedarwood, lemongrass, garlic, calendula, birch tar, rosemary, bergamot, lime, wintergreen, verbena, eucalyptus, peppermint, black pepper, and DEET 25%. Insects were exposed for a maximum of 2 min to each compound [63,64]. The paper arena was divided into 4 sides: 3 non-stimuli “NS”, including the closest to the oil and the horizontal sides of the sheet (1, 2 and 4 on Figure 1), and 1 stimulus zone “S”, which was the furthest from the oil (3 on Figure 1). As soon as an individual reached one of the edges, it was removed from the sheet and its position was accounted for. The number of insects reaching the “S” zone was recorded for each treatment, and the number of *T. castaneum* remaining was simply added to the “NS” distribution. Individuals that did not respond were excluded from the analysis. The number of insects in the “S” or” NS” zones was analyzed using a chi-square (χ^2^) contingency test in GraphPad Prism. The theorical distribution was 25% per side, or effectively 25% for the “S” and 75% for the remaining nonstimulus sides, because the release arena has four edges but only one stimulus edge, so to calculate the “proportion that are expected to exit under the null hypothesis” it would be ¼ = 25% and to calculate the proportion that should exit on the nonstimulus edges would be ¾ = 75%.

##### Group Exposure

Following the outline of the individual exposure above, 20 *T. castaneum* were tested at once to investigate the response as a group. In addition, the arena was surrounded by a sticky trap made of tape, placed to avoid escapes. Oils tested here include citronella, lime, wintergreen, eucalyptus, bergamot, black pepper, peppermint, and verbena. Insects reaching each side were trapped on the tape and counted at the end of the experiment. Each condition (compound) was repeated in triplicate. Oils that showed the most promising results after the first three trials were repeated six times. The distribution obtained for the oils tested were compared to a theoretical distribution (25% for S and 75% NS sides). A total of n = 3–6 replicates *T. castaneum* cohorts were tested per treatment combination.

#### 2.3.2. Large Wind Tunnel

A laminar flow wind tunnel (94.5 × 73 × 80.5 cm L:H:W; Appendix A) was used to assess the upwind orientation of *T. castaneum* to each stimulus. The wind tunnel consisted of an electric fan which pushed ambient air at 0.39–0.41 m/s, passing first through two metal sieves to straighten turbulent airflow and then an activated carbon filter to purify the air. For each oil, 10 µL was pipetted on a filter paper and was placed 9 cm away from both the last sieve in a 100 mm × 15 mm plastic Petri dish and 9.5 cm upwind of a 21.6 cm × 27.9 cm paper release arena. The same distances previously described for the small wind tunnel were maintained in this setup. The large wind tunnel was housed in a walk-in environmental chamber (Percival Scientific Inc., Perry, IA, USA) at 28 °C ± 2 °C and 65% ± 5% RH. Illumination was provided by fluorescent tubes lengthwise and widthwise along the wind tunnel directly overhead in an omnidirectional diffuser box (69.9 cm × 121.9 cm W:L). The essential oils tested in this experiment that were undiluted were calendula, cedarwood, citronella, garlic, lime, rosemary, and verbena. Potential positive controls of N,N-Diethyl-m-toluamide (DEET) 25% diluted in acetone, methylbenzoquinone (MBQ), and transfluthrin (TF) were also tested. Concentrations of MBQ and TF were prepared in acetone at the concentrations indicated above. These oils were selected from results of the small wind tunnel experiment as having the most significant deviation from the distribution obtained with water as negative control. These oils had the most significant *p*-values, although two more oils were also significant but could not be tested (least significant response in the previous experiment, lemongrass and birch tar). Two negative controls were tested for comparison: water for undiluted compounds and acetone for acetone-diluted compounds. Individual *T. castaneum*, n = 30 for each compound, were exposed successively. Individuals were released in the middle of the arena. As soon as an individual exited one of the edges, it was removed from the sheet and its position was recorded. The number of insects reaching the “S” (e.g., stimulus edge) zone was recorded for each treatment. Simultaneously, the sheet of paper was virtually divided into two zones of equal surface “A” (away from the repellent source) and “B” (close to the tunnel), and this information was recorded as well (Figure 1). Each cohort was given 2 min to respond to the stimulus. Adults that did not exit were treated as non-responders and were excluded from the analysis. Each release arena was changed between trials to prevent contamination by trace chemical stimuli deposited by beetles or the various treatments, and the door to the environmental chamber was opened and vented with a fan between treatments. A smoke test confirmed that the odor from the Petri dish reached the release arena. The number of insects that chose S or NS or “A” vs. “B” were analyzed separately using a χ^2^ contingency test in GraphPad Prism 10.3.0.

#### 2.3.3. Individual Behavioral Tracking with Ethovision

The behavior of red flour beetles was tracked with the software Ethovision XT 13.0 (Noldus, Wageningen, The Netherlands) [65]. Preliminary tests showed that the oils saturate the air in small (60 mm in diameter) and standard size (100 mm diameter) Petri dishes and that *T. castaneum* will run in circles, making it difficult to accurately measure their response (unpublished data, Appendix A). Thus, square cell culture Petri dishes, 24.1 cm × 24.1 cm × 2 cm (Nunc, Thermo Fisher Scientific, Waltham, MA, USA), were used instead to provide a larger area to quantify movements. The square dishes were lined with parafilm to provide a grip for the beetles and were placed in a dark chamber with a red light to avoid phototaxis. For each treatment, which consisted of 2 µL of the pure compound placed at a corner of the dish directly on the parafilm lining, 20 *T. castaneum* were observed for 10 min and multiple variables were compared between 5 oils: bergamot, citronella, eucalyptus, verbena, wintergreen, and a water control. Each treatment combination consisted of n = 20 replicates, and the orientation of the starting point was rotated 4 times to avoid external influences. The results were analyzed with a one-way ANOVA followed by Dunnett’s multiple comparison test using GraphPad Prism 10.3.0 (La Jolla, CA, USA).

#### 2.3.4. Taxis Release–Recapture Experiments

Plastic bins of 75 cm in length, 48.5 cm in width, and 35 cm in height were divided into 5 equally wide zones of 15 cm. The bottom (inside) of each bin was rubbed with sandpaper to provide traction to the insects, and bins were washed with soap and water, then rinsed, and then dried with Kim wipes (VWR, Arlington Heights, IL, USA) between replicates. Fifty *T. castaneum* were placed in the middle (neutral) zone where they would be allowed to acclimate and freely roam for 30 min. A perforated centrifuge tube, either empty or containing a cotton wick with 1 mL of the repellent to test, was added in the 1st zone (e.g., against one far side of the bin), which established a directional basis for recording decisions. The bins were placed in a walk-in environmental chamber with controlled temperatures at 28 °C ± 2 °C and humidity at 65% ± 5% RH. The oils tested were citronella, eucalyptus, lime, and verbena based on the results of the other assays (small wind tunnel assay and Ethovision behavioral study). Each condition was replicated 5 times, and the position of *T. castaneum* was recorded at 1, 8, and 24 h after the centrifuge tube of oil was placed in the bin. The number of insects per zone was analyzed with two different tests: a χ^2^ where the options were “close” (zone 1) or “far” (zones 2 to 5) to the area containing the repellent, and in a two-way ANOVA considering the effect of the zone number (proximity to the repellent or control) and of the treatments, with a follow up Tukey’s multiple comparison test. Each of the three timepoints were analyzed separately.

## 3. Results

### 3.1. Small Wind Tunnel

#### 3.1.1. Individual Exposure

One essential oil, cedarwood, exhibited a distribution significantly different from the theoretical distribution (χ^2^ = 4.59, df = 1, *p* < 0.016) (Figure 2), with significantly more *T. castaneum* exiting away from the repellent.

All other essential oils did not exhibit a significantly different distribution when compared using a χ^2^ test to the theoretical distribution. However, when comparing to water as a negative control, citronella, cedarwood, lemongrass, garlic, calendula, birch tar, and rosemary were significantly different from the control, with more *T. castaneum* moving away from each of these oils than for the control (Table 1, Figure 3).

#### 3.1.2. Group Exposure

We investigated the group behavior of a cohort of 20 *T. castaneum*. Three repellents exhibited distributions significantly different from the theoretical distribution: citronella, lime, and wintergreen (Table 1). For each of these oils, *T. castaneum* tended to move away from the oils more likely than by chance. The oils eliciting a significant repellent response were different from those identified when individual insects were exposed.

### 3.2. Large Wind Tunnel

None of the tested compounds produced a significant difference when compared to their respective controls (water or acetone) regarding the side (S vs. NS) where *T. castaneum* exited or the half (A vs. B) where the conspecifics exited (Table 2). TF and MBQ did not elicit repellency in *T. castaneum*.

### 3.3. Ethovision

Exposure to repellents significantly affected (Figure 4 and Table 3) mean distance moved (F = 3.95, df = 5, *p* = 0.0025), mean velocity (F = 4.502, df = 5, *p* < 0.001), mean time spent in the repellent zone (F = 6.448, df = 5, *p* < 0.0001), and mean distance from the repellent zone (F = 4.639, df = 5, *p* = 0.0007), but not angular (turning) velocity (F = 0.223, df = 5, *p* = 0.9519) of *T. castaneum*. The Dunnett’s multiple comparison results obtained showed that time spent in the repellent zone decreased significantly for all treatments compared to the control (Figure 4). Citronella and eucalyptus showed a significant increase in distance from the repellent zone compared to the control. Only eucalyptus showed a significant increase in the mean distance moved and the mean velocity by *T. castaneum*.

### 3.4. Taxis Release–Recapture Experiments

The number of insects close or far in the control compared to oil–time treatments showed that 3 oil–time treatments deviated from the control (χ^2^-tests, α < 0.05, Table 4, Figure 5). Lime at both 8 h and 24 h and verbena at 24 h showed a higher number of individuals moving away from these oils. Analyses of zones between treatments (Table 5) showed that the number of *T. castaneum* per zone was significantly different between zones for citronella at 8 and 24 h, lime at 1 h, eucalyptus at 8 and 24 h, and for verbena after 1 h. However, the other comparisons were not significant between control and oils.

## 4. Discussion

In this study, we tested candidate essential oils for repellent effects on the red flour beetle *Tribolium castaneum* that could be implemented as additional tools in pest management programs for mitigation of post-harvest loss. Depending on the assay performed, the behavioral outcomes differed, which showed multiple interesting results. In the group exposure experiment, *T. castaneum* distribution after exposure to citronella, lime, and wintergreen oils showed a significantly higher number of conspecifics in the “S” zone than in the theoretical distribution expected, which suggests the potential repellency of these three oils. However, the other experiments did not fully confirm these observations, although individual *T. castaneum* exposed to citronella in the small wind tunnel showed some repellency compared to the water control. In the bin experiment, citronella did not significantly have an effect on the behavior of the beetles, but their number differed between zones compared to a control at 8 and 24 h. These patterns are consistent with prior work on citronella oil, as it appears to have repellent effects on some insects, but not others, and depends highly on the conditions of delivery [66,67,68,69,70,71].

In the small wind tunnel experiment, cedarwood was the only oil eliciting a response that could show a repellency of the red flour beetle compared to the theoretical distribution. The second comparison shows that potentially seven oils—citronella, cedarwood, lemongrass, garlic, calendula, birch tar, and rosemary—could be efficient repellents compared to the water control when individual beetle behavior was assessed.

To enhance our understanding of the small wind tunnel results, the large wind tunnel experiment added a “positive control” and split the arena in two halves in addition to the four sides. However, none of these two comparisons significantly manipulated the behavior of *T. castaneum*, which may suggest that the compounds tested are not positive controls (Storgard as an attractant, MBQ and TF as repellents) or no suitable repellents (essential oils) were tested for adult *T. castaneum*. Some prior work with these positive controls confirms the sometimes inconsistent response of *T. castaneum* to the same compound when evaluated with multiple assays [59,72,73]. However, discrepancies between the small wind tunnel experiment and the large wind tunnel experiment could be explained by the different air flow speed between the two apparatus, which may influence how fast *T. castaneum* are exposed to these compounds. There may also be high variability in the behavior of *T. castaneum*, which can be exacerbated with a new environment. Another caveat to these experiments is that they were conducted with mixed-sex adults, so it is possible other volatiles may have been present. Future work should partition experiments by sex. In addition, the large wind tunnel was more enclosed than the small wind tunnel, which may have concentrated the odors, which can become overwhelming for *T. castaneum* and may have resulted in their behavior changing, although the chamber was aired out between treatments to minimize this issue.

In the taxis bin experiment, lime oil appeared to repel red flour beetles from the area containing the oil after 8 h and after 24 h compared to the control. Differences in potency between these oils may explain the timeline observed. We also compared the number of insects in each zone and between the oils and corresponding controls. We found that, overall, the number of *T. castaneum* in zones differs at different timepoints for the different treatments. The timing of behavioral response has been shown to be an important factor when assessing repellency, and also provided alternative results in different settings with various insects. For instance, bergamot was assessed as a potential repellent of stored-product beetle pests, but this was not confirmed in the present study [74,75]. However, the Ethovision behavioral analysis also revealed that some previously tested oils induced different behavior than the water control, particularly bergamot.

Finally, measuring behavior in response to oil presence using Ethovision showed some changes in insect behavior, including avoidance of the repellent zones for all oils tested. Both time spent in the repellent zone and distance away from the repellent zone were significant for most of the oils tested, suggesting that the beetles may be actively avoiding the space around these oils. Beetles also tended to increase their distance traveled and speed for all oils tested except wintergreen, which demonstrates that the beetles sensed the volatiles in their environment and adjusted their overall behaviors, which was previously shown [25]).

Overall, this study shows the importance of individual versus group behavior, timing of application, and response of repellents, as well as air flow, in the determination of repellency (or attraction). For example, group size and by consequence density was shown to affect foraging by *R. dominica* and *T. castaneum* [67]. Additionally, the use of a video tracking system for behavioral studies like Ethovision provides more parameters to quantify instantaneous responses of insects when exposed to a potential repellent that non-automated experiments do not allow [65]. Some modifications could be added to these assays, such as other insects, food, and water to bring more realism to the ecosystem in these scenarios. Group behavior might be a more realistic parameter to consider, as opposed to individual response, when testing for repellents against *T. castaneum* in urban and food storage areas, since *T. castaneum* tends to be attracted to its conspecifics [73]. Additionally, using pure essential oils may have caused saturation of the airspace, and dilutions may need to be assessed for discriminating value, as shown in other publications [68,74,76,77].

For stored-product insect pest management strategies, new limitations on some traditional management methods for postharvest are being increasingly driven by consumers, regulatory agencies, and biological changes such as insecticide resistance. Therefore, tools supporting resilient integrated pest management (IPM) informed by insect behavior are urgently needed. Prior studies have highlighted the need for more behaviorally based research on the potential of essential oils for IPM in stored products, particularly in combination with packaging options [68,75,78]. This study shows the importance of considering alternative setups to confirm the results obtained in a variety of behavioral experiments and highlights that results can be contradictory. Additional assays can provide additional confidence in the outcome or whether the oil is not consistently repellent. The other biological effects of oils should be observed in further experiments, such as reduction in reproduction or mortality depending on application methods. Whilst this paper presents the challenges faced by research in this area, it also provides data that furthers knowledge in these key directions of research that will support US and global food security and nutrition tool development.

## Figures and Tables

**Figure 1 insects-15-00626-f001:**
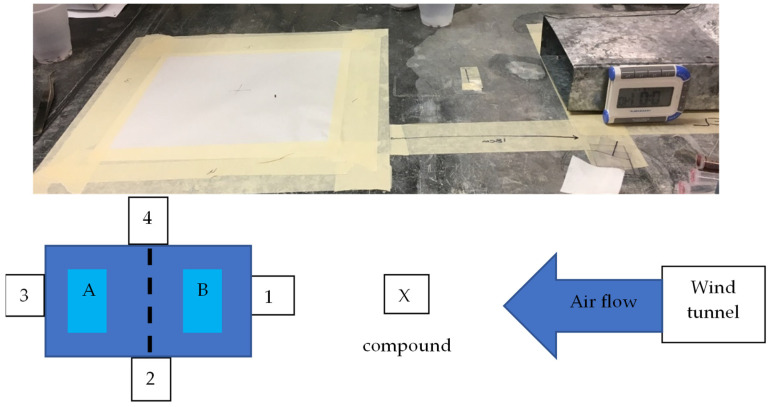
Experimental setup of the repellent test in the wind tunnel. The small wind tunnel area is pictured at the top, with a schematic of the principle for the two wind tunnel experiments at the bottom. The oil is placed on location “X”, and the beetle is placed in the center between A and B of the arena. Sides 1, 2, and 4 are non-stimulus sides, and side 3 is the stimulus zone furthest away from the oil.

**Figure 2 insects-15-00626-f002:**
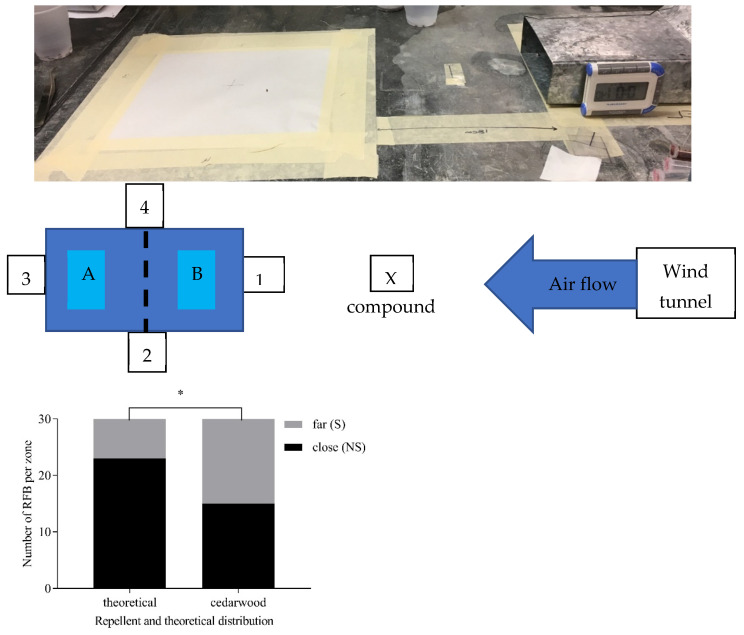
Comparison of the significantly different distribution of cedarwood exposure to a theoretical distribution in the small wind tunnel individual exposure experiment. Significance level: 0.01 < *p* ≤ 0.05: *.

**Figure 3 insects-15-00626-f003:**
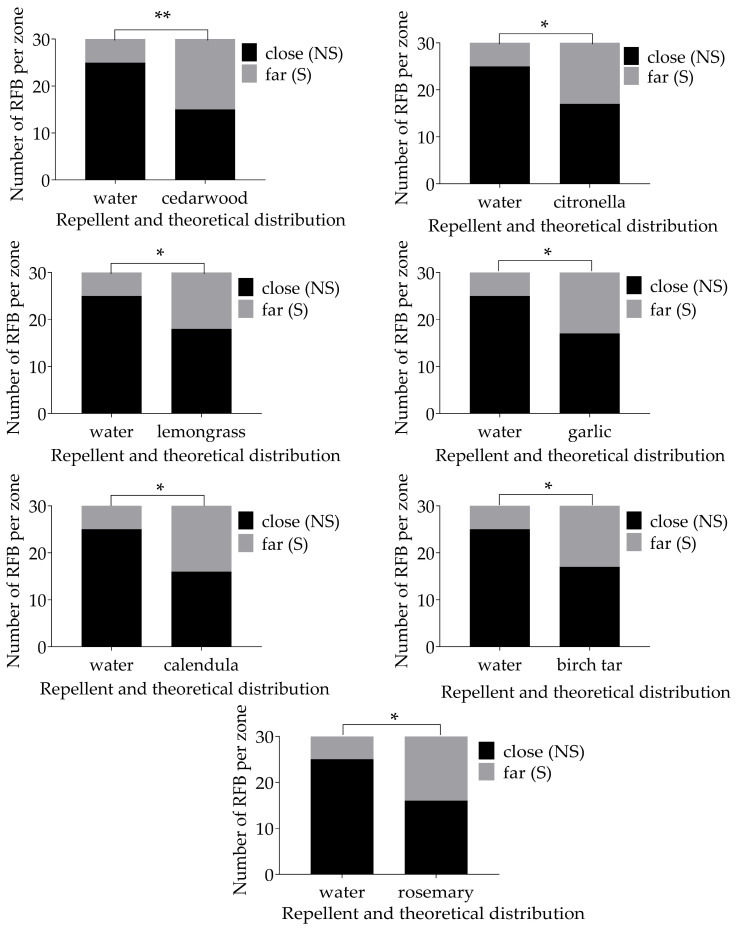
Comparisons of the distributions obtained after exposure to the seven oils differing from the control distribution in the small wind tunnel individual exposure experiment. * or ** denotes significant differences between preferences between the water and the oil showing a repellency response, 0.01 < *p* ≤ 0.05: *; 0.001 < *p* ≤ 0.01: **.

**Figure 4 insects-15-00626-f004:**
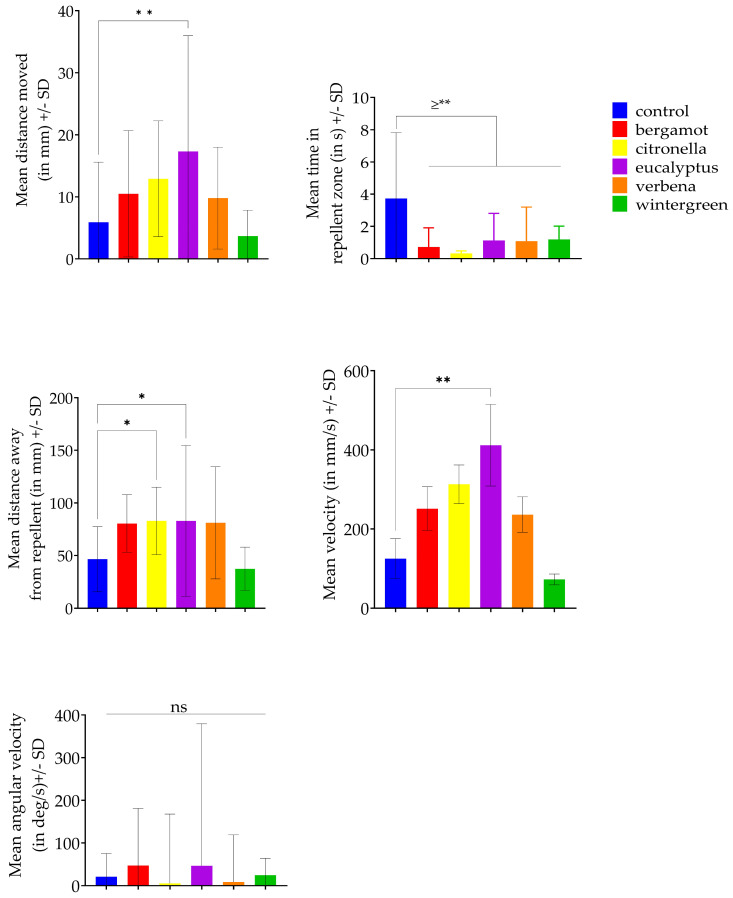
Ethovision parameters tested with ANOVA and Dunnet’s multiple comparison. Mean distance moved and velocity were different between eucalyptus and the control. Insects exposed to all treatments spent less time in the repellent zone compared to the control. Citronella and eucalyptus showed a significant increase in distance from the repellent zone compared to the control. No significant difference in angular velocity was detected between control and treatments. *p* > 0.05: ns; 0.01 < *p* ≤ 0.05: *; 0.001 < *p* ≤ 0.01: **. “≥**” is a summary of all the significance levels, in this case all *p*-values were 0.001 or lower.

**Figure 5 insects-15-00626-f005:**
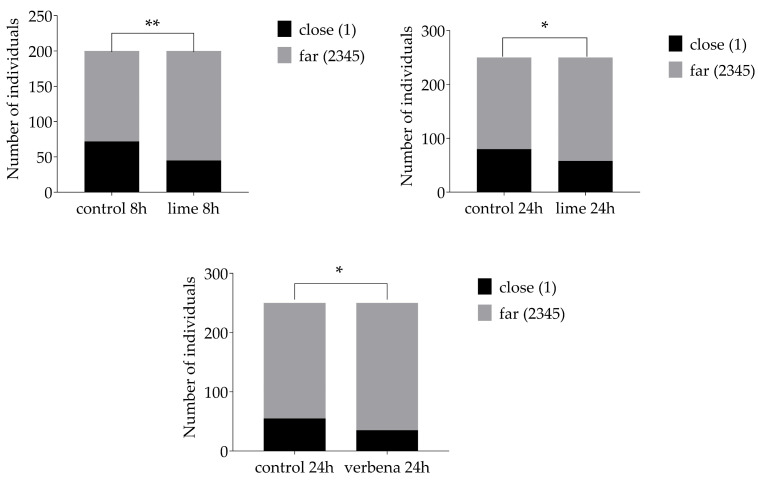
Distribution of red flour beetles in taxis bins, close or far from the repellent, significantly differing from the control distribution at different timepoints. Numbers between parentheses represent the zone numbers, close = 1 or far = 2, 3, 4, 5. 0.01 < *p* ≤ 0.05: *; 0.001 < *p* ≤ 0.01: **.

**Table 1 insects-15-00626-t001:** Calculated χ^2^ results for the oils, with distributions significantly differing from the control (water) in the small wind tunnel individual exposure experiment on the left and group exposure on the right. df = 1. *p* > 0.05: ns; 0.01 < *p* ≤ 0.05: *; 0.001 < *p* ≤ 0.01: **; *p* < 0.0001: ****.

	Oils	χ^2^	*p*-Value	Signif. Level
Comparison to a water control—individuals	citronella	5.079	0.024	*
	cedarwood	7.5	0.003	**
	lemongrass	4.022	0.045	*
	garlic	5.079	0.024	*
	calendula	6.239	0.013	*
	birch tar	5.079	0.024	*
	rosemary	6.239	0.013	*
	bergamot	0.8838	0.3472	ns
	lime	0.4167	0.5186	ns
	wintergreen	0.1113	0.7386	ns
	verbena	3.068	0.0798	ns
	eucalyptus	1.491	0.2221	ns
	peppermint	0.4167	0.5186	ns
	black pepper	2.222	0.136	ns
	DEET 25%	1.491	0.2221	ns
Comparison to a water control—group	citronella	17.01	<0.0001	****
	lime	24.9	<0.0001	****
	wintergreen	6.81	0.009	**
	eucalyptus	1.263	0.2611	ns
	bergamot	1.263	0.2611	ns
	black pepper	1.313	0.2518	ns
	peppermint	0.04547	0.8311	ns
	verbena	0.4301	0.5119	ns

**Table 2 insects-15-00626-t002:** Results of the large wind tunnel exposure experiment using *T. castaneum* for the two comparison types (S vs. NS, or A vs. B). The χ^2^, degree of freedom, and *p*-value are provided for each comparison in the large wind tunnel experiment. * The exposure to Storgard only had 20 replicates, whereas all the other treatments had 30 replicates. Df = 1, *p* > 0.05: ns, all were ns.

Comparison	χ^2^ S vs. NS	*p*-Value S vs. NS	χ^2^ A vs. B	*p*-Value A vs. B
citronella	1.176	0.2781	0.3175	0.5731
lime	0.2182	0.6404	0.6932	0.4051
verbena	1.92	0.1659	0.6932	0.4051
DEET 25%	0.1617	0.6876	0.08889	0.7656
cedarwood	0.5769	0.2238	0.6932	0.2025
garlic	1.92	0.1659	1.2	0.2733
calendula	0.1617	0.6876	0.8838	0.3472
rosemary	1.176	0.2781	0	>0.9999
transfluthrin 1 mg/mL	0.0982	0.754	0.07326	0.7866
methylbenzoquinone 40 mg/mL	0.1113	0.7386	0.3	0.5839
Storgard *	0	>0.9999	2.836	0.0922

**Table 3 insects-15-00626-t003:** Results of the Dunnett’s multiple comparison test (comparing all treatments to the control) in the Ethovision setup. 0.01 < *p* ≤ 0.05: *; 0.001 < *p* ≤ 0.01: **; 0.0001 < *p* ≤ 0.001: ***; *p* < 0.0001: ****.

Variable	Oils Significantly Different from the Control	Df	Mean Difference with the Control	Adjusted *p*-Value	Significance Level
Mean distance moved	Eucalyptus	112	−11.41	0.0061	**
Mean velocity	Eucalyptus	101	−286.4	0.0036	**
Mean time spent in the repellent zone	Bergamot	106	3.008	0.0001	***
	Citronella	106	3.405	<0.0001	****
	Eucalyptus	106	2.598	0.0012	**
	Verbena	106	2.645	0.0014	**
	Wintergreen	106	2.54	0.0011	**
Mean distance from the repellent zone	Citronella	112	−36.35	0.0398	*
	Eucalyptus	112	−36.37	0.0363	*

**Table 4 insects-15-00626-t004:** Results of the χ^2^ analysis of the taxis bin data compared to a negative control. Df = 1 for all, *p* > 0.05: ns; 0.01 < *p* ≤ 0.05: *; 0.001 < *p* ≤ 0.01: **.

	Citronella			Lime			Eucalyptus			Verbena		
	1 h	8 h	24 h	1 h	8 h	24 h	1 h	8 h	24 h	1 h	8 h	24 h
χ^2^	0.497	2.375	1.382	0.010	8.807	4.844	0.010	0.010	0.145	0.488	3.391	5.420
*p*-value	0.481	0.123	0.240	0.922	0.003	0.028	0.919	0.921	0.703	0.485	0.066	0.020
Level	ns	ns	ns	ns	**	*	ns	ns	ns	ns	ns	*

**Table 5 insects-15-00626-t005:** Results of the ANOVA analysis per zone of the taxis bin data compared to a negative control. Df = 4 for zones and interactions, Df = 1 for treatment, *p* > 0.05: ns; 0.01 < *p* ≤ 0.05: *; 0.001 < *p* ≤ 0.01: **.

	Citronella			Lime			Eucalyptus			Verbena		
	1 h	8 h	24 h	1 h	8 h	24 h	1 h	8 h	24 h	1 h	8 h	24 h
Zones												
F	1.382	13.18	7.057	5.566	1.351	2.354	2.03	8.817	3.812	4.607	0.436	0.9729
*p*-value	0.257	<0.0001	0.0002	0.0012	0.271	0.070	0.109	<0.0001	0.010	0.004	0.782	0.4331
Level	ns	**	**	**	ns	ns	ns	**	*	**	ns	ns
Treatment												
F	0	0	0	0	2.535	0	0	0	0	0	2.04 × 10^−29^	0
*p*-value	>0.9999	>0.9999	>0.9999	>0.9999	0.120	>0.9999	>0.9999	>0.9999	>0.9999	>0.9999	>0.9999	>0.9999
Level	ns	ns	ns	ns	ns	ns	ns	ns	ns	ns	ns	ns
Interaction												
F	0.372	1.771	0.337	0.170	0.683	1.561	0.305	0.498	1.707	0.838	1.629	2.440
*p*-value	0.827	0.154	0.851	0.952	0.608	0.203	0.873	0.738	0.168	0.509	0.186	0.0625
Level	ns	ns	ns	ns	ns	ns	ns	ns	ns	ns	ns	ns

## Data Availability

The data presented in this study are available on request from the corresponding author at the following link: https://doi.org/10.15482/USDA.ADC/25343395.v1.

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
