# Peer review of "Challenges in Assessing Repellency via the Behavioral Response by the Global Pest Tribolium castaneum to Protect Stored Grains"

_insects, 2024, doi:10.3390/insects15080626_

Round 1

Reviewer 1 Report

Comments and Suggestions for Authors

The authors presented research on the repellency of different essential oils as a tool against stored product insect species. The research design is well described, and the results are clear and properly explained. Although, I found some minor issues that need to be addressed before the paper could be suitable for publishing. 

Line 62 Reference No 11, is not up to date, I recommend using a more recent one

Section Materials and Methods: Was the population of T. castaneum beetles used in the experiment mix gender? If yes, please indicate that.

Give authority for all mentioned plant species, for example, Citrus bergamia (Rissol); Cymbopogon flexuosus (Jowitt ex Bor)…

Line 246 Please use Italic style for T. castaneum (throughout the section Results)

Section Discussion: Can you explain why some oils caused better repellency in a group than in an individual exposure?

Author Response

Reviewer Comments & Responses:

Reviewer 1:

Comment 1: The authors presented research on the repellence of different essential oils as a tool against stored product insect species. The research design is well described, and the results are clear and properly explained. Although, I found some minor issues that need to be addressed before the paper could be suitable for publishing. 

Response 1: We appreciate the comments and the time taken by the reviewer, thank you. We intend to respond and address the issues here, please see below.

Comment 2: Line 62 Reference No 11, is not up to date, I recommend using a more recent one.

Response 2: Done, this has been replaced by: Buzby, J.C.; Farah-Wells, H.; Hyman, J. The Estimated Amount, Value, and Calories of Postharvest Food Losses at the Retail and Consumer Levels in the United States. 2014, USDA, Economic Research Service, Bulletin No. 121.

Comment 3: Section Materials and Methods: Was the population of T. castaneum beetles used in the experiment mix gender? If yes, please indicate that.

Response 3: Done. “Mixed-sex” was added to the M&M. 

Comment 4. Give authority for all mentioned plant species, for example, Citrus bergamia (Rissol); Cymbopogon flexuosus (Jowitt ex Bor)…

Response 4: Done. These have all been added.

Comment 5: Line 246 Please use Italic style for T. castaneum (throughout the section Results)

Resonse 5: Done. All instances of T. castaneum have been italicized.

Comment 6: Section Discussion: Can you explain why some oils caused better repellency in a group than in an individual exposure?

Response 6: It could be that minor conspecific cues are required to prime response to the repellent. T. castaneum has a complex communication system, including both aggregation pheromones and anti-aggregation pheromones. This is usually in response to density-mediated cues (e.g. Ponce et al. 2024). As a result, we have added the following to the discussion:  “For example, group size and by consequence density was shown to affect foraging by R. dominica and T. castaneum [65].” And, added the following citation:

Ponce, M.A., Ranabhat, S., Bruce, A., Van Winkle, T., Campbell, J.F. and Morrison III, W.R., 2024. Density-mediated foraging behavioral responses of Rhyzopertha dominica (Coleoptera: Bostrichidae) and Tribolium castaneum (Coleoptera: Tenebrionidae). Scientific Reports, 14(1), p.12259.

Reviewer 2 Report

Comments and Suggestions for Authors

Major comments:

-Title: It seems very general. It mixes up, first: a method for assessing repellence; second: IPM tools; and third: pest species. Please rewrite the title so that it better reflects the content of the MS.

-Simple summary: As indicated in the journal guidelines, this section should be written in a "not entirely scientific" form; in the sense that the problem addressed, and results obtained can be understood by a general audience. Please rewrite.

-Keywords: Do not include those already used in the title. Normally, the indexing of a paper is conducted by the title and the keywords. Please review and change those that are repeated and are also informative, e.g., Coleoptera.

-M&M:

-In general, this section needs major restructuring to make it possible to understand the various bioassays conducted. In addition, it is necessary to provide more information as indicated below.

- Lines 144-147: Give a brief description of the trials conducted and to be described in the following subsections, e.g., X trials were conducted, the first two with small wind tunnel exposing adults individually or in groups, etc. The order of the trials in sections 2.3.3. and 2.3.4 is not understood. Are they prior to or complementary to those conducted with wind tunnels? Please explain and/or reorder the sections.

-In addition, if the Authors do not wish to subsequently repeat the experimental designs, data used and/or the statistical analysis of the results, indicate this here in full, e.g., indicate the type of statistical analysis conducted, not just the statistical package used. Alternatively, write the above in each section of M&M.

-Figure 1: It should be localized in the text of the MS closest to its citation in the same and not in Results. Also, explain in the caption that it represents the zones (A, B, 1, 2, 3, and 4).

- Section 2.3.4. Please title more science based. Do the authors want to refer to "still-air olfactometers" (e.g., Roberts et al., 2023. DOI: 10.1111/eea.13351). Apply in the rest of the MS text. Please explain, the description of the equipment used is not understood. Is there a figure and/or photograph in the text?

Discussion:

-The Discussion does not present a structure that is considered adequate. It seems more logical to first discuss the bioassays with isolated insects and, subsequently, those conducted with groups of insects.

-Lines 134-135 in M&M: the compound is mentioned: methyblenzoquinone o MBQ, “a compound naturally released by T. castaneum in response to crowding [60]”. However, in Results and, subsequently, in the Discussion, the possible effect of semiochemicals associated with T. castaneum male adult are not mentioned; these compounds could have more important effects on the results. Specifically: “Tribolium males produce an aggregation pheromone that attracts both sexes [4,8-dimethyldecanal (DMD)]” (Fedina & Lewis, 2008, DOI: 10.1111/j.1469-185X.2008.00037.x10.1111/j.1469-185X.2008.00037.x)

Would it not have been better to use males and females separately in the bioassays? Since the percentage of sexes in bioassays with groups of adults is not known.

As in other insect Orders, it is not possible to sex adults by morphological characters associated with sex, but it is possible to sex adults in the pupal stage (e.g., Hinton (1949). Secondary sexual characters of Tribolium. Nature, 149: 500-501).

Do the authors think it would have been better to study the response of sex-differentiated adults? Could there have been effects and/or interference with male aggregation pheromones and volatile compounds in the assays? Please justify.

Minor comments:

-Supplementary Figure 1. Please specify the light sources and their position in the wind tunnel.

-In general, the position of the light sources in the different olfactometers and/or wind tunnels are not well described. Nor if diffusing screens were used for such illumination.

Author Response

Reviewer 2:

Major comments:

Comment 1: -Title: It seems very general. It mixes up, first: a method for assessing repellence; second: IPM tools; and third: pest species. Please rewrite the title so that it better reflects the content of the MS.

Response 1: The original manuscript title reads “Challenges in assessing repellency as an integrated pest management tool to protect stored grains, using the global pest Tribolium castaneum”

Following consideration of the reviewers comments we have altered the title to “Challenges in assessing repellency via the behavioral response by the global pest Tribolium castaneum to protect stored grains

Comment 2: -Simple summary: As indicated in the journal guidelines, this section should be written in a "not entirely scientific" form; in the sense that the problem addressed, and results obtained can be understood by a general audience. Please rewrite.

Response 2: Done. We have rewritten for a more general audience.

Comment 3: -Keywords: Do not include those already used in the title. Normally, the indexing of a paper is conducted by the title and the keywords. Please review and change those that are repeated and are also informative, e.g., Coleoptera.

Response 3:  Done, the keywords have been revised as follows: “Wind tunnel; Ethovision; IPM; semiochemicals; Coleoptera; essential oils”.

Comment 4: -M&M: -In general, this section needs major restructuring to make it possible to understand the various bioassays conducted. In addition, it is necessary to provide more information as indicated below.

Response 4: Done. We have supplied additional details throughout the materials and methods section to add clarification.

Comment 5: Lines 144-147: Give a brief description of the trials conducted and to be described in the following subsections, e.g., X trials were conducted, the first two with small wind tunnel exposing adults individually or in groups, etc. The order of the trials in sections 2.3.3. and 2.3.4 is not understood. Are they prior to or complementary to those conducted with wind tunnels? Please explain and/or reorder the sections.

Response 5: Done. We have clarified that these assays were done in a sequential order and are thus presented that way.

Comment 6: In addition, if the Authors do not wish to subsequently repeat the experimental designs, data used and/or the statistical analysis of the results, indicate this here in full, e.g., indicate the type of statistical analysis conducted, not just the statistical package used. Alternatively, write the above in each section of M&M.

Response 6: Done. We have clarified that the statistics are unique to each assays’ design.

Comment 7: Figure 1: It should be localized in the text of the MS closest to its citation in the same and not in Results. Also, explain in the caption that it represents the zones (A, B, 1, 2, 3, and 4).

Response 7: Done. We have moved the figure and clarified the caption.

Comment 8: Section 2.3.4. Please title more science based. Do the authors want to refer to "still-air olfactometers" (e.g., Roberts et al., 2023. DOI: 10.1111/eea.13351). Apply in the rest of the MS text. Please explain, the description of the equipment used is not understood. Is there a figure and/or photograph in the text?

Response 8: Done, title has been relabeled as follows, “Taxis Release-Recapture Experiments”. We have eliminated reference to still-air olfactometers.

Comment 9: Discussion: The Discussion does not present a structure that is considered adequate. It seems more logical to first discuss the bioassays with isolated insects and, subsequently, those conducted with groups of insects.

Response 9: The goal of this organization of the discussion is to highlight the most influential results or oils with the greatest impact on beetle behavior. We also want to highlight the combination of the results of all of the experiments and how all of the different assays confirm no particular clear repellency effects of the oils tested here. Separating the results based on bioassays individually would not provide a full explanation and important caveats to the entire story of repellency tested here.

Comment 10: Lines 134-135 in M&M: the compound is mentioned: methyblenzoquinone o MBQ, “a compound naturally released by T. castaneum in response to crowding [60]”. However, in Results and, subsequently, in the Discussion, the possible effect of semiochemicals associated with T. castaneum male adult are not mentioned; these compounds could have more important effects on the results. Specifically: “Tribolium males produce an aggregation pheromone that attracts both sexes [4,8-dimethyldecanal (DMD)]” (Fedina & Lewis, 2008, DOI: 10.1111/j.1469-185X.2008.00037.x10.1111/j.1469-185X.2008.00037.x)

Response 10: We add a caveat about the possible missed sex effects of the MBQ tested. In addition, we do acknowledge that the MBQ and other pheromones have had inconsistent responses in T. castaneum (line 403) and so these controls should continue to be reevaluated.

Comment 11: Would it not have been better to use males and females separately in the bioassays? Since the percentage of sexes in bioassays with groups of adults is not known. As in other insect Orders, it is not possible to sex adults by morphological characters associated with sex, but it is possible to sex adults in the pupal stage (e.g., Hinton (1949). Secondary sexual characters of Tribolium. Nature, 149: 500-501).

Response 11: Adult T. castaneum are possible to sex based on setiferous patch on male but not female, but this requires much more handling time of adults that could affect behavioral response to compounds and behavior in assays. In order to minimize handling time of adults, high replicate size was used and mixed-sex populations were tested. In many cases with density-related cues, there is not a pronounced sex-based difference, unlike with sex pheromones.

Comment 12: Do the authors think it would have been better to study the response of sex-differentiated adults? Could there have been effects and/or interference with male aggregation pheromones and volatile compounds in the assays? Please justify.

Response 12: In hindsight, it would have been good to at least keep track of sex, and then block experiments by sex to rule out the effect of sex as affecting the Ethovision and group-based wind tunnel experiments. This is a future direction that has been incorporated in the discussion as follows: “Another caveat to these experiments is that they were conducted with mixed-sex adults, so it is possible other volatiles may have been present. Future work should partition experiments by sex.”

Minor comments:

Comment 13: Supplementary Figure 1. Please specify the light sources and their position in the wind tunnel. In general, the position of the light sources in the different olfactometers and/or wind tunnels are not well described. Nor if diffusing screens were used for such illumination.

Response 13: Light sources were from directly overhead along the length of the wind tunnel, were diffused, and omnidirectional. To reflect this, we have added the following in the M&M: “Illumination was provided by fluorescent tubes lengthwise and widthwise along the wind tunnel directly overhead in an omnidirectional diffuser box (69.9 cm × 121.9 cm W:L).” In addition, Supplementary Figure 1 has been revised to reflect the location of the light source.

Reviewer 3 Report

Comments and Suggestions for Authors

The manuscript by Rault et al. provides a comprehensive assessment of the repellency of various essential oils against the red flour beetle, Tribolium castaneum, a significant pest in stored grain ecosystems, aiming to contribute to integrated pest management strategies for stored grain protection. Despite not identifying significant repellent compounds, the study provides valuable data and insights to guide future research in this area. There are several areas where clarification and expansion are needed to improve the manuscript.

1.     In the “Introduction” section, a more detailed discussion on previous repellency studies related to T. castaneum would strengthen the background, providing a clearer foundation for the study’s objectives.

2.     In the “Materials and Methods” section, you could enhance the explanation of why specific essential oils were chosen for testing repellency by discussing previous research that indicates their effectiveness against T. castaneum.

3.     It would be improved by including a summary table that lists all the essential oils tested along with their purity.

4.     In the wind tunnel assays, you could explain the rationality for choosing distilled water as a negative control. I would like to know if distilled water has an attractive effect on T. castaneum. Also, it’s better to explain the scientificity of theoretical distribution (25% for S and 75% NS sides).

5.     How do the findings of this study compare with previous research on essential oil repellency against T. castaneum

Author Response

Reviewer 3:

Comment 1: The manuscript by Rault et al. provides a comprehensive assessment of the repellency of various essential oils against the red flour beetle, Tribolium castaneum, a significant pest in stored grain ecosystems, aiming to contribute to integrated pest management strategies for stored grain protection. Despite not identifying significant repellent compounds, the study provides valuable data and insights to guide future research in this area. There are several areas where clarification and expansion are needed to improve the manuscript.

Response 1: We appreciate the comments and the time taken by the reviewer, thank you.

Comment 2:  In the “Introduction” section, a more detailed discussion on previous repellency studies related to T. castaneum would strengthen the background, providing a clearer foundation for the study’s objectives.

Response 2:

Comment 3: In the “Materials and Methods” section, you could enhance the explanation of why specific essential oils were chosen for testing repellency by discussing previous research that indicates their effectiveness against T. castaneum.

Response 3: Regarding justification of our choices of essential oils based on literature. If an effect was known for these oils then we have a citation in the text, but our choice was based on product development criteria/ potential success as a new tool for IPM, for example, palatable taste for the consumer down the line in grain if they were successful repellents.

Comment 4: It would be improved by including a summary table that lists all the essential oils tested along with their purity.

Response 4: The following 100% pure therapeutic grade essential oils tested were purchased from Piping Rock (Ronkonkoma, NY); we tested bergamot (Citrus bergamia Risso), citronella (Cymbopogon winterianus Jowitt ex Bor), eucalyptus (Eucalyptus globulus Labill.), lime (Citrus aurantifolia [Christm.] Swingle), verbena (Verbena officinalis L.), wintergreen (Gaultheria procumbens L.), cedarwood (Juniperus ashei J. Buchholz), lemongrass (Cymbopogon flexuosus [Nees ex Steud.] W.Watson), calendula (Calendula officinalis L.), birch tar (Botula alba Roth), rosemary (Rosemarinus officinalis Spenn.), black pepper (Piper nigrum L.), and peppermint (Mentha piperita L.). Additionally, garlic (Allium sativum L.) 100% pure therapeutic grade essential oil was purchased from Healing Solutions (Phoenix, Arizona) on Amazon in September 2018

Comment 5: In the wind tunnel assays, you could explain the rationality for choosing distilled water as a negative control. I would like to know if distilled water has an attractive effect on T. castaneum. Also, it’s better to explain the scientificity of theoretical distribution (25% for S and 75% NS sides).

Response 5: Distilled water was the solvent in some cases and has no odor in and of itself. Thus, it was used as a negative control. For example, in an assessment of attraction in an y-tube olfactometer, humidifing channels with water was found to have no effect on attraction by T. castaneum (Stevenson et al. 2017, Journal of Economic Entomology, Walking Responses of Tribolium castaneum (Coleoptera: Tenebrionidae) to Its Aggregation Pheromone and Odors of Wheat Infestations: https://academic.oup.com/jee/article/110/3/1351/3061484). In addition, the release arena had four edges, and three of these edges were classified as non-stimulus ones, while only one was a stimulus edge, thus the distribution for the stimulus edge is 1/1+3=25% and for the nonstimulus edges it is 3/1+3=75%. As a result, we have added the following to the M&M: “This is because the release arena has four edges, but only one stimulus edge, so to calculate the proportion that are expected to exit under the null hypothesis” it would be ¼=25%, and to calculate the proportion that should exit on the nonstimulus edges, it is ¾ = 75%.”

Comment 6: How do the findings of this study compare with previous research on essential oil repellency against T. castaneum

Response 6: I believe this question is addressed here (please see below excerpt from the introduction), and the key thrust of the paper identifies that methodology is paramount in defining if a repellent is going to be as effective as expected. Users regularly report mixed experiences with essential oils as repellents in most fields of use, the work in this paper begins to elucidate the complexity of insect behaviors when exposed and also can shed light on why users of products may see varied and sometimes disappointing results. We hope that this paper sheds light on this and can support research going forward that can begin to de-mystify the effects of essential oils to some extent.

 “One way that repellency can be achieved is through the use of essentials oils and plant extracts, which have been found to have potential for repellency and some level of toxicity [44,50–52]. Essential oils are considered eco-friendly approaches for pest management, mainly in juxtaposition to synthetic insecticides [33,53,54] and have shown potential for synergy with phosphine for T. castaneum [55] and as fumigants for other stored-product insects [25] or with pyrethroids in other insects, such as mosquitoes [56–58].

However, repellency testing requires assessing behavior, which can be difficult and subject to a large amount of environmental variation [59]. In this study, we set up multiple types of experiments, aiming at identifying candidate repellents that could be used in IPM programs to manage T. castaneum populations in stored product warehouses and small farms. Although some of the tested essential oils showed promise, results were inconsistent from one experiment to the next. This study shows how critical data interpretation is in the context of adding additional tools to an IPM program in stored products.”

Round 2

Reviewer 2 Report

Comments and Suggestions for Authors

Figure 1: Does not exist? A photograph and a diagram appear without caption without further explanation. Indicate (A) and (B) and the descriptions.

Author Response

Comment: Figure 1: Does not exist? A photograph and a diagram appear without caption without further explanation. Indicate (A) and (B) and the descriptions.

Response: The description appears as below in the re submitted manuscript, it seems correctly described and labeled, please let us know if there is further issue with the manuscript, for example this explanation is not showing up for the reviewer, but it is from our side. 

Many thanks! 

Figure 1. Experimental setup of the repellent test in the wind tunnel. The small wind tunnel area is pictured at the top with a schematic of the principle for the two wind tunnel experiments at the bottom. The oil is placed on location “X” and the beetle is placed in the center between A and B of the arena. Sides 1 , 2, and 4 are non-stimulus sides and side 3 is the stimulus zone furthest away from the oil.

Reviewer 3 Report

Comments and Suggestions for Authors

The revised version of the manuscript has been improved, and the authors have addressed all my concerns.  

Author Response

Comment: The revised version of the manuscript has been improved, and the authors have addressed all my concerns

Response: Many thanks for your time and support in reviewing this manuscript.